# Advancements in Nanosystems for Ocular Drug Delivery: A Focus on Pediatric Retinoblastoma

**DOI:** 10.3390/molecules29102263

**Published:** 2024-05-11

**Authors:** Kevin Y. Wu, Xingao C. Wang, Maude Anderson, Simon D. Tran

**Affiliations:** 1Department of Surgery, Division of Ophthalmology, University of Sherbrooke, Sherbrooke, QC J1G 2E8, Canada; yang.wu@usherbrooke.ca (K.Y.W.);; 2Faculty of Medicine and Health Sciences, McGill University, Montreal, QC H3T 1J4, Canada; 3Faculty of Dental Medicine and Oral Health Sciences, McGill University, Montreal, QC H3A 1G1, Canada

**Keywords:** retinoblastoma, drug delivery systems (DDS), nanosystems, nanoparticles, nanotechnology, nanomedicine, ocular drug delivery, pediatric ocular disease, retinoblastoma

## Abstract

The eye’s complex anatomical structures present formidable barriers to effective drug delivery across a range of ocular diseases, from anterior to posterior segment pathologies. Emerging as a promising solution to these challenges, nanotechnology-based platforms—including but not limited to liposomes, dendrimers, and micelles—have shown the potential to revolutionize ophthalmic therapeutics. These nanocarriers enhance drug bioavailability, increase residence time in targeted ocular tissues, and offer precise, localized delivery, minimizing systemic side effects. Focusing on pediatric ophthalmology, particularly on retinoblastoma, this review delves into the recent advancements in functionalized nanosystems for drug delivery. Covering the literature from 2017 to 2023, it comprehensively examines these nanocarriers’ potential impact on transforming the treatment landscape for retinoblastoma. The review highlights the critical role of these platforms in overcoming the unique pediatric eye barriers, thus enhancing treatment efficacy. It underscores the necessity for ongoing research to realize the full clinical potential of these innovative drug delivery systems in pediatric ophthalmology.

## 1. Introduction

In the realm of pediatric oncology, retinoblastoma emerges as the foremost intraocular malignancy, presenting unique challenges in its treatment due to the delicate context of its pediatric patient base and the intricacies of ocular drug delivery. Conventional treatment modalities, including enucleation, chemotherapy, and radiotherapy, while effective to varying degrees, are fraught with limitations, such as systemic toxicity, recurrence, and long-term side effects, highlighting the urgent need for more targeted and personalized therapeutic strategies. In this context, the burgeoning field of nanotechnology offers a promising horizon. This review article delves into the current landscape of retinoblastoma treatment challenges and showcases the revolutionary potential of nanosystems in enhancing ocular drug delivery. Through a comprehensive exploration of various nanoparticle types—including gold, mesoporous silica, iron, polymeric, and lipid nanoparticles—this article elucidates their application in overcoming the biological barriers to drug delivery in the eye, potentially offering targeted therapy with reduced side effects and improved efficacy. Covering the literature from 2017 to 2023, this article comprehensively examines these nanocarriers’ potential impact on transforming the treatment landscape for pediatric ocular diseases.

## 2. Overview of Retinoblastoma

### 2.1. Epidemiology

Retinoblastoma is recognized as the most prevalent primary intraocular malignancy among children, securing its position as the second most common of such tumors across all age groups, only surpassed by uveal melanoma. This malignancy manifests with a frequency ranging from 1 in every 14,000 to 20,000 live births, translating to an estimated 250–300 new cases emerging annually within the United States alone. It exhibits no preference regarding gender or ethnicity, affecting both sexes and all races indiscriminately. Notably, in 30–40% of instances, retinoblastoma presents bilaterally [1]. A significant majority, approximately 90%, of retinoblastoma cases are identified in children younger than three years of age.

The age at diagnosis of retinoblastoma is intricately linked to both the presence of a family history of the disease and its bilateral or unilateral presentation. Children with a known familial history of retinoblastoma are typically diagnosed at an average age of 8 months. In contrast, those affected by the disease in both eyes, or bilaterally, are usually diagnosed at around 12 months of age. Meanwhile, patients exhibiting unilateral disease, where the tumor is confined to one eye, are often diagnosed at a later stage, around 24 months [1].

This variation in diagnosis timing highlights the influence of genetic predispositions and the scope of the disease’s manifestation. Furthermore, the incidence of retinoblastoma on a global scale exhibits a stark variation, with up to a fiftyfold difference primarily attributed to the variability in birth rates across different regions. Registries have recorded the highest incidences of retinoblastoma in African countries, indicating a geographical disparity in the prevalence of this condition [2].

### 2.2. Etiological Insights and Risk Factors

Retinoblastoma emerges due to detrimental alterations in the RB1 gene, specifically located on the long arm of chromosome 13 at the 13q14 locus [3]. The malignancy develops when mutations occur in both copies of the RB1 gene. In bilateral cases, which comprise a minority of occurrences, there is a 98% likelihood of the mutations being of germline origin. While familial retinoblastoma is rare, accounting for 5% of all cases, the vast majority of cases, or 95%, are sporadic. Within these sporadic cases, 60% are unilateral with no germline mutation detected, leaving the remainder exhibiting germline mutations and the potential for multiple tumor formations.

Heritable retinoblastoma is characterized by a mutation in one RB1 gene allele across all body cells. The disease progresses to malignancy when the remaining allele undergoes mutation, potentially due to environmental factors. This condition results in bilateral and multifocal tumors, significantly increasing the risk of secondary non-ocular cancers (e.g., pineoblastoma, osteosarcoma, soft tissue sarcoma, melanoma) [4]. The base risk of developing a secondary cancer is 6%, with this risk increasing fivefold if external beam radiation therapy is applied to the primary tumor.

Conversely, non-heritable retinoblastoma, which usually presents unilaterally, does not follow a pattern of genetic transmission and thus does not elevate the risk of secondary non-ocular cancers. Identified in patients with no family history of the disease, this type implicates a 1% risk for both the siblings and offspring of affected individuals. Significantly, non-heritable retinoblastomas constitute approximately 90% of unilateral cases [4].

### 2.3. Diagnostic Approaches and Clinical Presentation

#### 2.3.1. Initial Assessment and Clinical Signs

Diagnosis of retinoblastoma primarily relies on clinical assessment. Clinical signs and symptoms at the point of diagnosis vary depending on the tumor’s size and position. Leukocoria (a white reflection from the pupil) and strabismus (misalignment of the eyes) are the most frequently observed initial indicators of the disease [5]. Although less common, other symptoms, such as iris heterochromia, spontaneous hyphema, pseudo-hypopyon, and signs of ocular inflammation, may also be present (Figure 1). On occasion, smaller tumors might be detected during routine eye examinations, with complaints regarding vision being rare, largely because the condition predominantly affects children of pre-school age [6].

#### 2.3.2. Development and Variants

Retinoblastoma emerges as a translucent, gray-to-white tumor within the retina, supported by enlarged, twisted vessels. As it progresses, the tumor calcifies, adopting a distinctive chalky appearance. It exhibits two primary growth patterns: exophytic, developing beneath the retina and potentially causing widespread retinal detachment, and endophytic, growing on the retinal surface toward the vitreous cavity. Endophytic tumors may obscure blood vessels and are prone to producing vitreous seeds, which can spread within the eye, potentially forming new tumors. These seeds can also migrate to the anterior chamber, leading to nodules on the iris or a pseudo-hypopyon. Such cases frequently encounter complications, such as secondary glaucoma and rubeosis iridis. Diffuse infiltrating retinoblastoma, a rarer form identified later in childhood, usually manifests unilaterally and complicates diagnosis due to the obscured retina from dense vitreous cells, often being mistaken for uveitis with an unknown origin [6].

#### 2.3.3. Diagnostic Imaging and Evaluation

MRI is pivotal in diagnosing retinoblastoma due to its superior soft-tissue resolution and non-radiative nature. These modalities help identify the characteristic calcifications and evaluate the optic nerve, orbits, and brain. Systemic metastasis assessments, including bone marrow and lumbar puncture, are generally not recommended unless there are neurological symptoms or signs of extraocular extension [7]. Additionally, examining the parents and siblings for signs of retinoblastoma or retinocytoma is crucial to identify a hereditary predisposition.

#### 2.3.4. Metastasis and Extraocular Extension

In developed countries, it is uncommon for patients with retinoblastoma to have metastasis or intracranial extension at the time of diagnosis. When metastasis does occur, it typically involves the abdominal viscera, brain, bones, lymph nodes, and spinal cord, often as a result of the tumor spreading through the optic nerve or choroid. Growth of the tumor into the orbit can cause proptosis, and its invasion into the anterior chamber can promote lymphatic dissemination, which may be detected through palpable lymph nodes [8].

#### 2.3.5. Classification

The management of retinoblastoma has evolved significantly, with primary local and/or systemic chemotherapy now being preferred over external beam radiotherapy. This shift has led to the widespread adoption of the International Classification System for Intraocular Retinoblastoma globally. Within this framework, the classification of eyes is based on tumor size, the presence of subretinal fluid, and the degree of vitreous and subretinal seeding. The system categorizes eyes from Group A (most salvageable) to Group E (least salvageable) with chemotherapy. Group E encompasses cases with anterior chamber involvement, neovascular glaucoma, vitreous hemorrhage, and/or necrosis, typically deemed unsalvageable [9].

### 2.4. Prognosis and Management Strategies

In industrialized nations, the prognosis is favorable, with survival rates above 95% for intraocular cases. However, this rate significantly declines to below 50% upon extraocular extension. Thus, the treatment protocols prioritize, in descending order, the preservation of life, the eye, and vision. The contemporary management of intraocular retinoblastoma integrates various modalities, including enucleation, systemic and local chemotherapy, laser and cryotherapy, along with external beam and plaque brachytherapy. For cases exhibiting metastatic spread, a combination of intensive chemotherapy, radiation therapy, and bone marrow transplantation is adopted. A multi-disciplinary team approach, involving ocular oncologists, pediatric ophthalmologists, pediatric oncologists, and radiation oncologists, is essential for optimizing treatment outcomes [10].

## 3. Challenges in Treating Retinoblastoma

### 3.1. Enucleation

Enucleation stands as a cornerstone treatment for retinoblastoma, offering a complete surgical removal of the tumor in the majority of cases. This procedure is typically deemed suitable under certain conditions, including when the tumor occupies over half of the eye, when there is suspicion of orbital or optic nerve involvement, when the anterior segment is affected, in the presence of neovascular glaucoma, or when the vision potential of the affected eye is severely compromised. The surgical technique aims to maximize the length of the optic nerve removed, ideally exceeding 10 mm [11].

### 3.2. Laser and Cryotherapy

The treatment of retinoblastoma frequently involves the use of the 810 nm infrared diode laser, highly regarded for its precision in targeting retinal tumors. This technique can be administered as a standalone treatment or as an adjunct to chemotherapy, offering flexibility in managing the disease [12].

For smaller tumors, with an apical thickness up to 3 mm, cryotherapy is preferred. It employs a triple freeze–thaw technique under direct observation, distinguishing its application based on tumor location: laser photoablation for posterior tumors and cryoablation for anterior ones. Both methods require multiple sessions and continuous monitoring to address tumor recurrence or treatment-related complications effectively [13].

### 3.3. Radiotherapy

External beam radiation therapy (EBRT) was a widely popular method for treating RB in the past. It is a local treatment delivering external radiation beams to the target tissue. A consequence of this treatment modality is its potential to induce secondary tumors. Wong et al. (1997) found that genetic predisposition to hereditary RB and radiation increase the risk of secondary malignancy [14]. Abramson et al. (1998) suggested that the long-term effect of radiation treatment depends strongly on the age at first exposure to EBRT, and it is thus not recommended before 12 months of age [15]. Though effective, its potential major complications cause this treatment to be of last resort. Yousef et al. (2020) found that EBRT was able to control RB eyes after the failure of combined chemotherapy and focal therapy [16]. Today, this technique is recommended in patients who are not candidates for safer therapeutic modalities [17].

### 3.4. Chemotherapy

Chemotherapy is a great alternative to EBRT, as it spares the risks of secondary malignancy. Systemic chemotherapy is not shown to be effective when administered alone and can cause important consequences, such as hearing deficits, neutropenia, and infection [18]. Intra-arterial delivery of chemotherapy can cause neutropenia and the need for blood transfusion, although to a lesser extent than systemic delivery. Furthermore, the invasive nature of arterial delivery can result in systemic side effects, such as transient ischemic attacks and strokes [19]. Additionally, vascular effects, such as vitreous hemorrhage, arterial occlusion, and retinal detachment, can occur [16,19]. Although the cause remains unknown, some suggest that it is a drug-specific side effect. Steinle et al. (2012) demonstrated vascular toxicity of melphalan and carboplatin (CRB) [20], and histological changes supporting such toxicity were shown by Tse et al. (2013) [21]. Intra-arterial injections are also limited by the blood–retinal barrier (BRB) and the avascular nature of the vitreous; thus, repeated injections are required to express adequate vitreous drug concentrations [22,23] (Figure 2). Intravitreal drug delivery is the preferred method for treating vitreous seeds. This approach allows for direct delivery of drugs to the vitreous and increased local bioavailability. However, this raises the question of potential toxicity to the eye and surrounding tissues due to high concentrations within the vitreous. Yousef et al. (2021) concluded that aftereffects at standard doses are not uncommon, with 59% developing side effects, with retinal toxicity and cataracts being the leading results [24]. Intravitreal injections also often require multiple injections, increasing the chances of endophthalmitis. Nanoparticles (NPs) as drug delivery systems can address the toxic complications, drug stability, and physiological barriers currently hindering chemotherapy injections. Table 1 provides a summary of treatment types and respective challenges discussed in this section.

## 4. Nano-Based Strategies

### 4.1. Inorganic Nanoparticles

#### 4.1.1. Gold Nanoparticles (GNPs)

GNPs have a range of possible applications, from diagnostic tools to therapeutic delivery, including low toxicity and high surface area to volume ratio, enabling high loading capacity and delivery to the retina [25]. Yan et al. (2021) took advantage of their high surface area to volume ratio to coat a gold(I)-thiol-peptide with polyacryl sulfhydryl imidazole (PSI). PSI coating of the GNPs resulted in a lower Zeta potential at pH 7.4, falling from 37 mV to 28 mV, and the NPs’ spherical shape remained unchanged. Peptide-derived molecular glue was embedded to target Murine Double Minute X (MDMX) degradation, which allows for the restoration of p53 and p73, two tumor suppressor genes. Additionally, PSI coating enabled the nanoparticle to be pH-responsive, only releasing its content in a tumor microenvironment, which is normally more acidic than normal tissue [26,27]. In vitro studies on WERI-RB1 cells demonstrated a dose-dependent degradation of MDMX with a DC50 of 154 nM. In a patient-derived xenograft (PDX) model of pancreatic carcinoma measuring 100 ± 25 mm^3^, this NP inhibited tumor growth by 85.7% within 10 days, and downregulation of MDMX was seen on histological analysis. Furthermore, no weight change, difference in blood biochemical index, or pathological findings were observed following treatment [26,27]. Kalmodia et al. (2017) loaded GNPs with anti-human double minute 2 (anti-HDM2) and studied their effect on Y79 cells. These GNPs were spherical, 32–122 nm in size, with a Zeta potential of −11.4; the hydrodynamic diameter (HD) and polydispersity index (PI) were 66.72 and 0.303, respectively. Their experiment on Y79 cells showed a statistically significant decrease in cells in the G1 phase following treatment, suggesting a mechanism of G2/M phase arrest and p53-mediated apoptosis [28]. These two studies illustrate the wide variety of structures that GNPs can take to target the same problem of retinoblastoma. Darviot et al. (2019) merged laser treatment and nanoparticles to reduce the energy required to destroy cancerous cell membranes via heat. Two types of GNPs were utilized: citrate-capped 100 nm and 80 nm 50:50 gold–silver. Interestingly, at 250 J cm^−2^, the laser–GNP interaction resulted in up to 98% cell death in the vitreous phantom of 15 cP, mimicking the vitreous environment, compared to 82% in the culture environment, showing optimistic results for clinical application. However, one limitation is the lack of homogenous attachment of NPs to the cultured cells, since their interaction relies solely on electrostatic attraction [29]. Hence, there is opportunity to explore surface coatings targeting Y79 cells for future enhancement of the treatment modality. Similarly, Moradi et al. (2020) demonstrated the synergistic effect of ultrasonic hyperthermia and GNPs. The fabricated spherical GNPs had a diameter of around 60 nm, with a Z potential of +38.6 mV and a hydrodynamic diameter of 89 nm. In vitro studies on Y79 cells highlighted that cell viability of 50% was achieved after 9 min of hyperthermia alone, whereas it was achieved in 4.5 min with the presence of gold nanoparticles [30]. Moradi et al. (2020) followed up with another study investigating the synergistic effect of 20 Gy brachytherapy, hyperthermia, and GNPs in rabbit eye models inoculated with Y79 cells. These GNPs had similar properties, being spherical, 60 nm in size, with HD of 80 nm and Zeta potential of +38.6 mV. Though this combination therapy caused a significant reduction in tumor size, resulting in tumor growth inhibition of 85%, compared to their individual potentials, it also enhanced inflammation, scarring, and hemorrhage occurrences [31]. Thus, there is a need for investigation into the protocols addressing variable factors, such as the ideal temperature, length of treatment, and heat source, before delving into clinical studies.

#### 4.1.2. Mesoporous Silica Nanoparticles (MSNPs)

MSNPs have been investigated for drug delivery and imaging among others, attributable to their high surface, pore volume, drug-loading capacity, and ease in incorporating surface functionalization [32]. There is a substantial research focus on the efficient targeting of tumor cells. Folic acid receptors are significantly overexpressed in retinoblastoma cells, and targeting such receptors could benefit cellular uptake [33]. Hence, their high drug-loading capacity and availability for surface-targeting properties could benefit chemotherapeutic delivery to retinoblastoma cells. Qu et al. (2018) explored this concept by synthesizing folic acid conjugated MSNPs embedded with topotecan. The addition of surface folic acid (FA) not only had benefits in terms of cellular uptake but also enabled controlled drug release. The original spherical MSNP containing topotecan (TPT) had a size of 110 ± 2.54 nm and increased to 168.5 ± 1.65 nm following the addition of FA, but it kept its spherical shape. In vitro release pattern testing using a dialysis protocol highlighted the controlled release of MSNPs with FA receptors, as 60% of TPT was released after 24 h of incubation compared to 80% in MSNPs without FA receptors. In vivo testing in Y79-cell-bearing xenograft tumor models demonstrated the efficiency of folic acid in targeting tumor cells, which allowed for a reduction in tumor volume of ~300 mm^3^ compared to ~1000 mm^3^ in MNPs without FA receptors, which further supports the idea of increased cellular uptake due to the presence of FA receptors [34]. Using a similar approach of targeting retinoblastoma (RB) cells and in order to address carboplatin (CRB) side effects, such as neurological toxicities and thrombocytopenia, and limitations of physiological barriers, Qu et al. (2017) used MSNPs to effectively deliver the drug. Rather than depending on diffusion, this nanoparticle targeted epithelial cell adhesion molecule (EpCAM) receptors overexpressed in RB cells by attaching EpCAM to the surface of the MSNPs loaded with carboplatin (CRB). The nanoparticles were spherical and had a Zeta potential of −20.1 ± 2.15 mV and a size of ~150 nm. An IC_50_ value of 1.38 μg/mL compared to free CRB IC_50_ value of 3.26 μg/mL demonstrated its significantly higher cytotoxicity attributed to its higher cellular internalization due to ligand targeting. Further, in vitro testing on Y79 cells revealed that free CRB induced 15% apoptosis compared to 40% and 65% in CRB-containing MSNPs and those with EpCAM receptors, respectively [35,36]. Sodagar Taleghani et al. (2019) opted to design a stimuli-responsive and cancer-cell-targeting MSNPs to control and target drug release. Taking advantage of their unique biological need for above-normal glucose uptake and the higher extracellular pH of tumor tissues compared to healthy tissue, the MSNPs loaded with deferasirox and containing pH-responsive polyamidoamine (PAMAM) dendrimer and decorated with glucuronic acid had a Zeta potential of 5.3 mV. In in vitro studies, at pH 7.4, only 15% of deferasirox was released compared to 66% at pH 4.5, demonstrating the sensitivity of PAMAM dendrimers to pH. The viability of Y79 cells when induced with deferasirox-loaded sugar-decorated nanocarrier was around 39.85 ± 0.45% for a concentration of 200 μg/mL compared to 72.84 ± 1.29% for those without sugar conjugation, highlighting the benefits of sugar conjugation in the uptake of the drug via receptor-mediated endocytosis [37].

#### 4.1.3. Iron Nanoparticles

Iron NPs have a particular characteristic of being of superparamagnetic nature, allowing them to be excellent candidates in biomedical imaging as well as hyperthermic in nature in response to magnetic fields [38]. Demirci et al. (2019) used dextran-coated magnetic NPs in magnetic hyperthermia, where internalized nanoparticles are exposed to a magnetic field and convert the energy into heat, killing tumor cells, which are sensitive to heat. These NPs were not found to be toxic in the absence of a magnetic field and were selectively cytotoxic to Y79 RB cells instead of ARPE-19 cells upon magnetic induction. At a concentration of 0.75 mg/mL, 50% of Y79 cell death occurred compared to ARPE-19 cells, which had a viability of 88%. Moreover, apoptosis began less than 24 h after induction and involved both the mitochondrial cell death pathway and apoptotic gene expression [39]. Seydi et al. (2023) studied superparamagnetic iron oxide nanoparticles (SPIONs) targeting the mitochondria of Y79 RB cells. These SPIONs had an average particle size of 20–40 nm and a bulk density of 1.20 g/cm^3^. This resulted in mitochondrial swelling, a collapse in mitochondrial membrane potential (MMP), a decrease in succinate dehydrogenase (SDH), and an increase in caspase-3 and ROS, leading to an overall decline in cell viability and an IC_50_ of 250 nm in RB mitochondria [40]. Although the in vitro results are positive, the exact properties of such nanoparticles remain unknown, such as penetration capability in retinal tissue and the choroid [39]. Hence, it is important for in vivo experiments to explore potential ocular toxicity. Further, although the hyperthermic properties of iron NPs allow them to exert therapeutic effects alone, it would be interesting to explore the possible synergistic effects of hyperthermic reactions and chemotherapeutic drugs loaded in iron NPs.

#### 4.1.4. Silver Nanoparticles

Silver nanoparticles (AgNPs) are alternative vehicles for delivery and have been widely applied in the antimicrobial context due to their broad-spectrum antimicrobial activity. They also have great potential in anti-cancer treatment due to their low toxicity, magnetic properties, and surface area to volume ratio. Importantly, AgNPs alone tend to agglomerate due to their surface charges, hence the importance of surface coating, especially in the context of drug delivery through the vitreous—a size- and charge-sensitive, difficult environment to cross [41]. Turbinaria Ornata are algae containing Laminarin, a polysaccharide with anti-tumor and anti-apoptotic features among other biological activities [42,43]. Remya et al. (2017) first developed a technique for rapidly synthesizing biological AgNPs from Turbinaria Ornata extracted from brown seaweed. Biologically synthesized AgNPs have enhanced therapeutic effects compared to their chemical counterparts while being environmentally friendly and cost-efficient [44,45,46,47]. Mixing 10 mL of the natural extract with 90 mL of 1 mM aqueous silver nitrate (AgNO_3_) yielded polydispersed AgNps with a size of 22–32 nm and a Zeta potential of −28.7 mV. In vitro experimentation demonstrated beneficial cytotoxic and apoptosis effects on Y79 cells with good radical scavenging effects and IC_50_ of 10.5 μg/mL [46]. Remya et al. (2018) followed up with a similar study where the AgNPs were synthesized using 5 mL of laminarin and 95 mL of 1 mM AgNO3, which resulted in 21–34 nm uniform and spherical particles with IC_50_ of 16.45 μg/mL in Y79 cell lines. Furthermore, the mechanism involved breaking DNA double strands, leading to increased sub-G1 apoptotic cell growth and inhibiting proliferation [46,47]. The biosynthesis of these AgNPs using algae allows for an environmentally friendly, cost-effective synthesis method [48].

### 4.2. Organic Nanoparticles

#### 4.2.1. Polymeric Nanoparticles

Polymeric nanoparticles are composed of various polymers, with the most common being poly(lactic-co-glycolic acid) (PLGA) and polycaprolactone (PCL), which are approved by the FDA [49]. Furthermore, they are biocompatible and degradable, with high stability and solubility. Moreover, their structure allows for enhanced drug accumulation, which is ideal for loading and delivering intact drugs to the retina, enhancing their bioavailability at the site of action [50]. Godse et al. (2021) studied the role of etoposide-loaded, chitosan-coated, and galactose-conjugated PLGA NPs in Y79 cells in vitro. These spherical PLGA NPs had a Zeta potential of +25 mV, which is favorable for diffusion across the vitreous and penetration of the retina, and the size ranged from 150 to 160 nm. Galactose targeted RB cells through their overexpression of sugar receptors, allowing for a cellular uptake of 70.02% compared to 49% in PLGA NPs without galactose in Y79 cells. Chitosan stabilized the NPs, permitting sustained release of the drug for over 32 h compared to a complete release of the free drug after 6 h. The enhanced properties of this nanoparticle permitted increased cellular uptake, cytotoxicity, and apoptosis [51]. Silva et al. (2019) adopted a similar approach, where they evaluated the cytotoxic effects of oleanolic (OA) and ursolic (UA) acids loaded in PLGA NPs and demonstrated decreased drug cytotoxicity when loaded in nanoparticles. These nanoparticles had sizes ranging from 213.55 ± 1.60 nm to 217.98 ± 2.74 nm and Zeta potentials of −27.12 ± 0.27 mV and −26.85 ± 0.49 mV depending on the type of mixture of OA and UA being loaded—natural or synthetic, respectively. In vitro analysis of Y79 cells demonstrated cell viability of 15.43% when exposed to the free synthetic mixture of OA and UA compared to 65.62% with PLGA NPs. Advantageously, in vitro exposure to these nanoparticles in HepG2 and Caco-2, colon and hepatic cells, showed the absence of systemic toxicity, highlighting their potential as oral dosage forms [52]. Zhuang et al. (2021) exposed carboplatin-loaded, sodium alginate (SA) surface-modified PLGA NPs to Y79 cells in vitro, which highlighted the benefits of SA when it comes to cellular uptake and the pace of drug delivery. The PLGA NPs were round, with a mean size of 253.94 ± 66.53 nm and a Zeta potential of around −43.1 ± 8.1 mV. The burst release of CRB was examined at 24 h and was 18.86% ± 4.3% and 40.01% ± 8.2% for PLGA NPs with and without SA, respectively. Analysis of the inhibitory effect on Y79 cell viability showed that PLGA NPs with SA exhibited higher effects at all concentrations and times compared to SA-absent PLGA NPs. Their future incentives include in vivo topical administration of their engineered NP as a treatment comparable to intravitreal drug delivery and its side effects [53]. Based on these studies, there exists an endless possibility of combinations of types of polymers, drugs, and surface modification. The therapeutic results of each variable are yet to be understood. For instance, Sims et al. (2019) tested melphalan-loaded PLGA NPs with various surface modifiers, and only the Tet Methylcytosine Dioxygenase 1 (TET1) and Methoxy Polyethylene Glycol (MPG) groups demonstrated improved efficacy relative to unmodified NPs. MPG-modified NPs had a diameter of 129 ± 38 nm, and TET1 had a diameter of 123 ± 33 nm. Melphalan loading at 1 mg/mL saturated batches in MPG and TET1 NPs showed statistically significant differences: 135.7 ± 7.8 and 212.7 ± 4.5 μg melphalan per NP, respectively. The IC_50_ values of MPG and TET1 were 0.35 ± 0.06 and 0.23 ± 0.04 mg NP/mL, respectively, and they differed in binding patters, with MPG PLGA NPs having the highest level of cell internalization after 1.5 h compared to the TET1-modified group, which only increased after 24 h. These results demonstrate the impact of surface modifications on the changing physiology of NPs [23]. Likewise, when synthesizing a nanoparticle, it is important to consider the trade-offs between positively and negatively charged carriers, respectively, to improve internalization and distribution. Lastly, the size of NPs is an important factor when considering leakage through important biological barriers, which could cause systemic toxicity [51]. Hence, there exists a strong need for investigation into the optimized structures of nanoparticles and their modifiable factors to highlight the ideal combinations specific for treating retinoblastoma. Finally, chitosan-derived nanoparticles are a sub-type of polymeric NPs and have been investigated to overcome the administration challenges with natural chitosan, which is insoluble in neutral media. Comparative studies between N-trimethyl chitosan (TMC) and thiolated chitosan (TC)—the derivatives of chitosan—highlighted the superiority of thiolated chitosan nanoparticles (TCs-NPs). In fact, Delrish et al. (2021) believed that chitosan nanoparticles could address the barriers of effective intravitreal delivery of topotecan hydrochloride (TPH) in treating RB and aimed at comparing the effectiveness between topotecan-loaded N-trimethyl chitosan nanoparticles (TPH-TMCs-NPs) and TPH-TCs-NPs in addressing these obstacles, such as controlled drug delivery and poor cellular uptake. TPH-TMCs-NPs and TPH-TC-NPs were both spherical in shape but had different sizes (39 ± 3, 25 ± 2 nm), Zeta potentials (38 ± 3, 12 ± 2 mV), and EE (73.34 ± 2, 85.23 ± 2%). The difference in these properties reflected different outcomes in xenograft models intravitreally injected with Y79 cells, where TPH-TC-NPs exhibited 90% tumor necrosis compared to TPH-TMCs-NPs (20%) and free TPH (15%) It is hypothesized that TPH-TCs-NPs showed superior cellular uptake and cytotoxicity and allowed for penetration of the vitreous due to their small size, low anionic, and small Zeta potential (12 ± 2 mV), which allowed them to efficaciously penetrate the vitreous humor compared to the latter, which had a higher Zeta potential [54]. Following this discovery, Delrish et al. (2022) engineered carboxymethyl dextran (CMD)-TMCs-NPs with a surface charge of 29 ± 4.31 mV and CMD-TC-NPs with a charge of 11 ± 2.27 mV. The diameters of these chitosan NPs were 42 ± 4.23 and 34 ± 3.78 nm, respectively. They assessed the biodistribution of these NPs in xenograft mouse models previously injected with Y79 cells and highlighted once again the effect of surface charge of NPs on post-intravitreal diffusion to the retina. CMD-TMCs-NPs were completely immobilized due to their highly positive charge [55]. Recently, Ghassemi et al. (2023) combined all these findings to synthesize TPH-CMD-TCS-NPs. Combining TPH drug therapy, CMD for improved NP stability, and TCS as a vehicle for delivery, they were spherical in shape, with a size, PDI, Zeta potential, and IC_50_ of 30 ± 4 nm, 0.24 ± 0.03, 10 ± 3 mV, 40.40, respectively. In vivo studies in rabbit xenograft models demonstrated a significant decrease in rabbit tumor volume seven days after injection, highlighted by 91 ± 2% tumor necrosis. Additionally, no histological retinal deformities were found, suggesting an alternative anti-tumor treatment with low toxicity [56].

#### 4.2.2. Lipid Nanoparticles

Lipid-based NPs (LNPs) are appealing due to their biocompatibility, degradability, scale-up capacity, and possibility of delivering hydrophilic and lipophilic drugs specifically, such as etoposide—a lipophilic medication used as chemotherapy in treating retinoblastoma. There exist numerous kinds of lipid NPs, varying from liposome-based, solid-lipid-based (SLNs) to nanostructured lipid carriers (NCLs), which are composed of a mix of solid and liquid lipids [57]. Ahmad et al. (2019) highlighted the variable features of lipid NPs based on the nature of the lipid matrix, its concentration, and particle size. For example, lipid concentration plays an important role in entrapment efficiency (EE) and particle size. Based on this, they engineered an optimized LNP, with a particle size, polydispersity index (PDI), and EE of 239.43 ± 2.35 nm, 0.261 ± 0.001, and 80.96 ± 2.21%, respectively. In vitro drug release studies comparing free etoposide and etoposide loaded in the optimized lipid NP demonstrated biphasic and slow release in the latter—75.74% ± 1.56 after 48 h—compared to the near total release of the free drug after 16 h. This finding also supports the sustained vitreous concentration of etoposide following intravitreal administration of the lipid NP in enucleated mouse eyes. The NPs also allowed for sustained release of etoposide for at least seven days compared to a trace amount of the free drug after the second day [58]. Marathe et al. (2022) studied a α-tocopherol succinate (αTS)-based nanostructured lipid carriers (NCLs) to overcome the poor aqueous solubility of paclitaxel (PTX). This study highlighted the importance of considering the solid and liquid lipid ratio depending on the loaded drug and its properties, as this parameter can greatly influence the parameters of NLCs, such as size. Their optimized NLCs had a size, PDI, and Zeta potential of 186.2 ± 3.9 nm, 0.17 ± 0.03, and −33.2 ± 1.3 mV, respectively. Marathe et al. further developed on the importance of variables when constructing a NP by evaluating its stability over 60 days. NLCs containing polyethylene glycol (PEG) NLCs demonstrated higher stability in terms of particle size, PDI, and Zeta potential compared to pure NLCs, suggesting an alteration in the physiochemical properties of NLCs after PEGylation. Drug release reached a maximum of 40% and plateaued until the end of the study at 48 h compared to 67% for the PEG NLCs [59]. Both studies have underlined the potential lipid nanoparticle structural changes possible when altering the variables. When looking at a combination therapy of genetic therapy and drug delivery, Tabatabaei et al. (2019) fabricated spherical, switchable LNPs containing melphalan and miR-181a with a size of 171 ± 4 nm, Zeta potential of 24.5 ± 0.7 mV, and PDI of 0.13 ± 0.04. The switchable LNPs allowed for conformation change and endosomal escape at acidic pH. The EE of melphalan and miR-181a were 93.0% ± 0.2 and 97.5% ± 0.7, respectively. They concluded that the delivery of dual-loaded LNPs containing melphalan and miR-181a resulted in amplified effects compared to both being delivered separately. For instance, LNPs containing miR-181a alone resulted in 37% reduction in Y79 cells compared to 72% when delivering LNPs containing both miR-181a and melphalan in in vivo treatments of mouse vitreous [60]. Conversely, Gibson et al. (2020) manufactured switchable LNPs containing siRNA. Use of the extrusion method yielded LNPs with a size of 162 ± 1 nm, Zeta potential of 31.6 ± 1.0 mV, and PDI of 0.135 ± 0.03. These siRNA LNPs targeting survivin sequential to chemotherapy increased the therapeutic effects of melphalan and carboplatin, with survivin being resistant to the drugs before its downregulation. Further, survivin silencing before chemotherapy allows for retained effectiveness at a lower dose, reducing its potentially toxic effects. Indeed, survivin silencing allowed for melphalan and topotecan IC_50_ to be reduced by 77%. As an illustration, treatment of Y79 cells using 30 μM of CRB and 0.25 μM of topotecan following survivin pre-treatment was necessary for 50% cell viability compared with 191 μM and 0.5 μM, respectively, when used individually in the absence of miR-181a [61]. Finally, still on the topic of synergistic effects, Wang et al. (2023) investigated the co-delivery of melphalan and black phosphorus quantum dots (BPQDs) dually loaded in LNPs for photothermal therapy. These spherical LNPs were 82–180 nm in diameter, with a PDI of 0.15–0.16, EE of 40%, and 43.7% for melphalan and BPQDs, respectively. In vivo studies in mouse xenograft models inoculated with WERI-RB-1 cells revealed that co-loaded LNPs showed an inhibitory effect of 65% compared to melphalan alone (51%) on tumor proliferation. One interesting finding is that free BPQDs were more cytotoxic than those loaded in nanoparticles, which is optimal for mitigating their toxic side effects, such as nephrotoxicity. In fact, BPQDs alone induced the highest inhibitory effect, at 79%, but they showed the highest ki67 levels, which were reduced in co-loaded LNPs [62,63]. As mentioned previously, ongoing research to optimize the formulation of LNPs only or within a combination therapy is needed to better understand the characteristics and changes in the structure and function, which loading of different elements may cause. For example, as previously discussed, Tabatabaei et al. (2019) highlighted that incorporation of miR-181a reduced the surface charge of LNPs from 40.4 ± 8.2 to 24.5 ± 0.7 mV but maintained their cationic properties [60]. Dual loading of elements can also cause incompatibility or tissue toxicity, hence the importance of interaction analysis before delivery. Indeed, Gibson et al. (2020) showed that all drugs have different properties, and the targeting of survivin silencing did not enhance the therapeutic potential of Teniposide compared to other drugs, such as carboplatin, suggesting a different intracellular mechanism of this drug [61].

#### 4.2.3. Protein Nanoparticles

As with lipid nanoparticles, protein NPs can be generated using various building blocks, such as albumin and lipoprotein, and they have advantages in their preparation, where toxic chemicals can be omitted. They are also stable, cost-effective, and have high loading capacity [64]. Moreover, proteins are amphiphilic, allowing for interactions with hydrophilic and hydrophobic drugs, and their abundance in charged groups enables various chemical modifications, such as surface-modified targeting properties [65]. Therefore, their high loading capacity and selective targeting capabilities can allow for optimal cancerous cell uptake. In fact, cancer stem cells—which are found in both primary RB and Y79 cells—have chemoresistant properties, since they induce cell arrest, while chemotherapy targets proliferating cells [66,67,68,69]. Narayana et al. (2021) produced lactoferrin protein nanoparticles (LFNPs)—targeting the prominent lactoferrin (LF) receptors in tumor cells due to increased iron demand—to treat retinoblastoma and mitigate chemoresistance to carboplatin (CRB) and etoposide (ETP). LFNPs containing CRB (CRB-LFNPs) and those containing ETP (ETP-LFNPs) had dimensions and EE of 61.2 nm, 60%, and 45.15 nm, 38%. They found that LF-CRB and LF-ETP displayed increased cellular uptake, retention, and cytotoxicity compared to their free drug counterparts and that the release of NP content was highest at pH 6. As an illustration, the analysis of the drug effect on Y79 cell population yielded free CRB IC_50_ of 230.3 compared to 118.2 for CRB-LFNPs. Moreover, the free CRB uptake was 9.99 μg/10^6^, with drug retention at 15.46% after 48 h compared to CRB-LFNPs, where drug uptake and retention were higher (17.38 μg/10^6^ and 54.43%). Similar trends were observed when comparing free ETP and those loaded in LFNPs. Their experiments centered on cancer stem cells (CSCs) due to their characteristic feature of drug resistance. The increased cytotoxicity of these LFNPs on Y79 CSCs offers promising potential in targeting the drug resistance of CSCs. It is worth evaluating the possibility of dual drug loading or sequential NP delivery of both CPT and ETP for potential synergistic properties [65]. The optimal results in vitro may be translated to future in vivo experiments.

#### 4.2.4. Nanomicelles

These nanoparticles are structured with amphiphilic monomers, hydrophobic heads, and hydrophilic tails, permitting high hydrophobic drug loading, stability, and solubility [70]. Curcumin’s hydrophobic nature, poor bioavailability, and rapid degradation have pushed for the discovery of curcumin analog alternatives, such as 3,4-difluorobenzylidene curcumin (CDF), to address these issues [71]. Alsaab et al. (2017) constructed amphiphilic poly(styrene-co-maleic acid)-conjugated-folic-acid (SMA-FA) CDF nanomicelles to enhance the targeting ability of the hydrophobic drug in the hopes of reducing the therapeutic levels of chemotherapeutic agents, which often lead to toxicity when present at high levels. The spherical nanomicelle had a size, PDI, Zeta potential, and EE of 193.6 ± 20 nm, 0.175 ± 0.05, –7.12 ± 4 mV, and 75.98 ± 12%, respectively. The targeting formulation allowed for higher therapeutic activity in Y79 cells, as reflected by a smaller IC50 of 1.02 ± 0.55 µM compared to 4 ± 1.64 µM for non-targeting NPs (i.e., nanomicelles without folic acid). However, such trend was not found in WERI-RB1 cells due to their lower folate receptor expression. Nonetheless, these targeting nanomicelles result in increased cytotoxicity due to higher internalization of the drug. This delivery method can help address the conventional use of high concentrations of chemotherapeutics due to poor retinal uptake of the free drug [72]. These results could perhaps be translated to natural curcumin, addressing its degradation and bioavailability challenges. Celastrol, a Chinese herbal medicine, is a topic of interest within nanomicelles. This bioactive component has previously shown potential in chronic inflammatory or auto-immune diseases, as well as anti-cancer characteristics [73]. Li et al. (2020) sought to invent a novel method of treating angiogenesis-mediated retinoblastoma using celastrol-loaded polymeric nanomicelle (CNM) and addressing the poor water solubility of the herbal medicine. These CNMs had a spherical shape with a diameter of 48 nm, with a celastrol loading content of 7.36%, and they exhibited controlled release. In vitro studies on chick embryo chorioallantoic membrane assays (CAM) demonstrated positive results, inhibiting vascular formation in a concentration-dependent manner. Following intraperitoneal injection in xenograft mouse models with SO-RB50 cells, similar optimal results were found. Following recurrent administration of 27.2 mg/kg CNMs for 16 days, significant tumor suppression was observed. The average tumor weight was 61.40 ± 20.82 mg compared to the control, while the change in body weight was omitted. CNMs inhibit the vascular endothelial growth factor A (VEGF-A) pathway, which is critical in angiogenesis and prevents endothelial cell migration. This novel therapeutic model demonstrates potential in inhibiting angiogenesis-specific RB, with the opportunity to be studied in other angiogenesis-based ocular pathologies, such as retinopathy of prematurity [74]. Still on the topic of celastrol, Guo et al. (2021) set to design a redox-sensitive nanomicelle (RSNM) for tumor-targeted celastrol delivery. Their reduction-sensitive grafted copolymer poly[thioctic acid-grafted-poly(ethylene glycol)/(benzylamine)] nanomicelles were spherical, with a size of 70 nm and a celastrol loading of 8.51%. RSNMs and drug release occurred in lysosomal reduction environments. The cellular uptake in Y79 cells in vitro was rapid and was observed just 10 min following incubation of RSNMs, with celastrol migration to the nuclear contents of cells within 1 h and proliferation inhibition within 4 h. The controlled release in response to tumor microenvironments is beneficial for apoptosis due to its time-dependent properties. Further analysis concluded that celastrol does this through caspase-9 and caspase-3 activation within RB Y79 cells [75]. Studies comparing these reduction-sensitive nanomicelles and the targeting properties of (SMA-FA)-CDF could offer insights into the benefits and challenges of both enhancers. Furthermore, the revelation of celastrol’s effect on apoptotic and VEGF-A pathways highlights its various mechanisms of action, which can be used selectively.

Figure 3 illustrates the common methods of drug delivery, including those discussed in this section, such as intravitreal and topical delivery.

## 5. Clinical Barriers and Future Perspectives

### 5.1. Trends

#### 5.1.1. Smart Release of Nanoparticle Contents

As mentioned previously, the release of nanoparticle contents can be triggered by an environmental change, such as acidic pH within tumor cells [27,37,60]. This enables a targeted release, diminishing the potential risk of cytotoxicity in healthy cells. Likewise, Long et al. (2021) synthesized a green-light-responsive nanocarrier to overcome the inner blood–retinal barrier, which intravitreally injected drugs face (Figure 2). Clathrin-like trigonal molecules loaded with doxorubicin had a size of 99.04 nm, with a Zeta potential, PDI, and EE of −25.10 mV, 0.10–0.15, and 13.6%. In a constructed in vitro inner BRB model, they compared the amount of DOX found across the endothelial monolayer from DOX-loaded non-irradiated nanoparticles (20%) and irradiated nanoparticles (50%), highlighting the increased extravasation following the release of DOX. This is due to DOX having a smaller size and hydrophobicity, allowing it to cross the layer with ease compared to the NP complex. Additionally, free drug injections in WERI-RB-1-cell-bearing mice caused a 10% loss in body weight due to systemic toxicity, whereas this was not the case within the nanoparticle group following intravitreal injection. This method is unique, as it benefits from the properties of both NPs and DOX for optimal drug delivery. NPs are a vehicle for bringing DOX closer to its site of action, avoiding its rapid elimination via the intravenous route, until it reaches the BRB, where it is too large to cross the barrier. The drug is then freed from its vehicle to arrive at its site of action [76]. This therapeutic element demonstrates great potential even beyond retinoblastoma, as it can be applied to any disease site reachable by the green light but not by NPs. Nonetheless, there remains a need for more research studies on this subject, especially long-term toxicity follow-ups, to ensure safety before clinical experimentations.

#### 5.1.2. Surface Receptor Targeting Nanoparticles

It has been proven that retinoblastoma cells, among other cancerous cells, have modified physical properties and biological demands. Previous studies have synthesized lactoferrin nanoparticles to target the enhanced LF receptors on tumor cells, including Y79 cells. This characteristic is a result of increased iron demand in the rapidly dividing cancerous cells [77]. Folate-coated nanoparticles targeting RB cells through their increased folate receptors have also been widely studied, showing enhanced cellular uptake compared to their unconjugated counterparts [34,72]. Combining RB cells’ increased demand for iron and folate receptors, Sadri et al. engineered a nanoparticle with dual-targeting properties conjugated with folic acid and transferrin, which physiologically facilitates uptake in cancer cells. These oleic acid (OA)-coated superparamagnetic iron oxide NPs (SPIONs) were loaded with Vincristine (VCR), were spherical, and had the dimension, Zeta potential, loading capacity, and EE of 82.1 nm, 25.0 mV, 8.7%, and 87.15%. This study was able to highlight increased cellular uptake within Y79 RB cells compared to healthy ARPE-19 in vitro, suggesting the positive effect of dual receptors. This was supported by the IC50 value of the free drug (10.61) compared to that of the SPIONs (4.87) in Y79 cells [78]. However, there is a lack of direct comparison between mono-targeting NPs and dual-targeting NPs to illustrate the extent of their benefits. Finally, RB cells are known to express sugar receptors, with special affinity for galactose and mannose [79]. Although many studies have demonstrated positive effects in terms of the uptake and cytotoxicity of sugar-modified NPs, potential research comparing folate, sugar, and LF receptors can allow for a better understanding of their unique properties. In summary, these various targeting receptors all demonstrated optimistic results in cancerous cell targeting, which in turn increased cellular uptake and cytotoxicity while minimizing toxicity to other untargeted cells due to lower affinity.

#### 5.1.3. Multi-Functional Nanoparticles

Similar to a combination therapy—where systemic chemotherapy in addition to focal therapy is used to treat retinoblastoma—a nanoparticle can be engineered to have more than one mechanism of action at a tumor site. For example, Zou et al. (2022) experimented with tuftsin-loaded carbonized metal–organic framework (CMT) NPs with a Zeta potential of −12.1 mV and a size of 303.1 ± 48.53 nm. These CMT NPs were superparamagnetic in nature following carbonization, enabling tumor targeting under a magnetic field. This is different from the receptor targeting discussed previously, which is mainly used to bring the drug to its site of action and could present an alternative to complex environments lacking target-specific receptors. These NPs showed adequate and stable thermal conversion (27.08%), allowing for photothermal therapy (PTT). The loading of tuftsin enabled immunotherapeutic effects by inducing macrophage typing. The subsequent laser irradiation of CMT NPs allowed for tuftsin stimulation of in vitro macrophages close to the level of free-drug-treated cells. This was reflected by the high levels of positive CD16/32 cells (33.13 ± 0.23%) in the CMT NP group. In vivo inoculation of tumor-bearing mice highlighted the synergistic effect of PTT and immunotherapy compared to laser alone or a combination of laser and NPs in the absence of tuftsin. Laser and CMT NPs combined allowed for a gradual decrease in tumor size and resulted in the highest rate of volume suppression, illustrated by their low and gradually decreasing relative tumor volume curve in relation to their counterparts. In addition, CMT NPs enabled easy follow-up of treatment efficiency by acting as excellent dual-modal imaging contracts for photoacoustic (PA) and T2-weighted MRI imaging, with the PA signal peaking at 4 h in the tumor region and remaining in the tumor-bearing mouse models of Y79 for up to 24 h. In short, these nanoparticles are multi-functional, with both therapeutic and imaging properties [80]. Analogously, Zheng et al. (2022) demonstrated the benefits of selective-loading NPs with dual-function elements, allowing for a nanoparticle capable of synergistic PDT and PTT effects with ultrasound (US)/PA/MRI imaging potential while simultaneously enhancing cellular uptake due to dual-targeting surface receptors. Folate/magnetic dual-target cationic nanoliposomes with a size of 338.63 ± 10.90 nm and Zeta potential of 31.86 ± 3.49 mV were loaded with indocyanine green (ICG), perfluorohexane (PFH), and Fe_3_O_4_. In vitro studies on Y79 cells comparing mono- and dual-target NPs highlighted that dual-target NPs reached saturation first at 3 h compared to the other groups, which reached saturation of less than 50% by the same time. Importantly, ICG can display both PTT and PDT effects upon stimulation from the same laser, although its PDT effect is mild due to the hypoxic environment of tumors. The incorporation of PFH, which increases O_2_ in the hypoxic environment, enhanced the effect of PDT, a therapy dependent on the presence of oxygen. ROS produced by NPs containing PFH was 6.15 times higher than that produced by NPs in the absence of PFH. Furthermore, the in vivo performance of these nanoparticles in Y79-bearing mice showed the most optimal anti-cancer effect (*p* < 0.0001), as the presence of PFH allowed for the synergistic effect of PTT and PDT [81]. Mudigunda et al. (2022) created Palbociclib (PCB) and near-infrared (NIR)-dye-loaded hybrid (PLGA and PCL) polymeric NPs (PNPs) for the combined PCB and PTT effect. The NPs had a size of 176 ± 43.8, PDI of 0.240, and EE of 80 and 81.5 for PCB and IR, respectively. The in vitro effects on Y79 cells were optimistic, where PCB-IR-PNPs demonstrated the highest inhibitory activity, with 86.5 ± 2.3% cell death compared to free PCB and IR groups (62.5 ± 0.4 and 54.8 ± 6.9% viable cells). Finally, no acute (14 days) toxicity was documented [82]. Wu et al. (2018) utilized laser irradiation as a means of triggering gene transfection, and taking advantage of the targeting properties of FA, they built FA-surface-modified, PFP-, ICG-loaded, and plasmid-DNA-carrying LNPs with laser activation of gene therapy and imaging properties. They had a diameter, Zeta potential, and ICG EE of 328.4 ± 5.5 nm, 35.3 ± 1.9 mV, and 92.1 ± 1.3%. The cationic property of NPs allowed for enhanced plasmid binding of 22.7 ± 0.1 μg compared to 1.4 ± 0.4 μg in neutral NPs. The enhancement of gene transfection is believed to result from laser-mediated liquid-to-gas transition of the NPs, causing a cavitation effect and increased cellular permeability. In both in vitro (Y79) and in vivo (Y79) studies, FA-modified cationic LNPs exhibited enhanced therapeutic effects compared to FA-absent or neutral LNPs. In vivo PA imaging was superior in FA-conjugated cationic LNPs at all time points. At 12 h following injection, all PA values decreased, with FA-conjugated LNPs remaining the highest, slightly below 3 a.u. Finally, it is important to note that the in vivo studies were performed on nude mice with right hind limb subcutaneously implanted tumors, which does not reflect the microenvironment and physiological properties of the eye [83]. Compared to the previously discussed experiments, natural elements, such as Moringa oleifera (MO), display similar synergistic potential in PTT and cytotoxic effects, with potential in photoacoustic imaging due to the fluorescent properties of chlorophyll derivatives within MO. Indeed, Mudigunda et al. (2023) prepared MO and IR encapsulated PCL NPs, which were spherical, cationic, and 160–220 nm in size. PPT induced cytotoxicity in Y79 cells in vitro, causing cell death of 50% in the IR group in the absence of MO compared to 80% in the IR+MO group following light induction [84]. Finally, Wang et al. (2020) identified NPs with low intensity focused ultrasound therapy (LIFU), imaging, and immunotherapy potential using muramyl dipeptide (MDP) and perfluorobutane (PFP) encapsulated in magnetic hollow mesoporous gold nanocages conjugated with iron oxide NPs (AuNC−Fe_3_O_4_/MDP/PFP NPs). The former is an immunomodulator, and the latter provides an ultrasound contrast and synergistic effect with the presence of LIFU. They had a size of 159.3 nm, Zeta potential of −15.8 mV, and EE of 82.60 ± 2.07 and 27.11 ± 1.20 for MDP and Fe_3_O_4_, respectively. These NPs were found to release MDP rapidly following LIFU irradiation and activated dendritic cells (DCs). Moreover, the NPs were safe and had no toxic effect on the ARPE-19 or Y79 cells in the absence of LIFU at concentrations as high as 400 μg/mL. In vitro cytotoxic studies of LIFU indicated that the presence of PFP is integral to high treatment efficacy due to its role in acoustic energy aggregation following liquid–gas transition. In vivo testing on Y79-subcutaneous-flank-tumor-bearing mice highlighted the superiority of LIFU and AuNC−Fe_3_O_4_/MDP/PFP NPs, as they resulted in the highest apoptotic index and lowest proliferative index compared to their counterparts. Weight, hematoxylin and eosin (H&E) stain hematology parameters, and blood chemistry failed to detect toxicity [85]. There exist numerous approaches to achieving certain additional functions of NPs—for example, imaging using biological MO, CMT NPs, or ICG. Despite the increasing multi-functionality of NPs, with these complex combinations comes the risk of consequential interactions and potential toxicity to adjacent organs. 

### 5.2. Barriers of Nanoparticles in Retinoblastoma

#### 5.2.1. Long-Term Toxicity and Lack of Clinical Studies

Most studies on NPs in retinoblastoma have shown the absence of toxicity within nanoparticle-treated RB cells or tumors despite enhanced therapeutic effects. Histology was the main modality in evaluating toxicity in mouse models, with most reflecting impeccable retinas [56]. Mice were also weighed for potential signs of toxicity, identifying an absence of general weight loss [74,76,82]. Despite this, there is a lack of long-term follow-up for signs of toxicity. In fact, most studies only verify the absence of acute toxicity, which is observed for a few days to weeks. For example, two studies measured weight, blood samples, biochemistry tests, and H&E staining up to 14 days after injection of nanoparticles, but they were not followed up thereafter [85,86]. In vitro studies have also reported findings of an accumulation of CMT NPs and LNPs in the liver and spleen of tumor-bearing mice through fluorescence analysis, although this is hypothesized to be a normal physiological process of NP phagocytosis via the reticuloendothelial system present in both organs [80,87]. This uncertainty perhaps plays a part in hindering the advancement in RB NP research. Indeed, most studies were either performed in vitro on Y79 cells or in vivo in xenografted animal models, with some of the inoculated tumors inoculated in areas other than the eye, moving away from the ideal microenvironment. Long-term in vitro and in vivo follow-up will significantly solidify our understanding of the effects of these novel therapeutics before moving to clinical studies with RB patients.

#### 5.2.2. Exploration of Common Variables

The engineering of unique nanoparticles requires choosing the type of NP, conjugation, loading, embedding, and synthesis method, making the possibility of combinations endless. As mentioned previously, Delrish et al. (2022) demonstrated the impact of a molecular change on the nanoparticle’s overall function. Comparing TC (11 ± 2.27 mV) and TMC NPs (29 ± 4.31 mV), TMC NPs were unable to cross the vitreous to the retina due to the methylation of chitosan and overall positive charge, supporting the hypothesis that surface charge is the limiting factor relating to intravitreal injections reaching their designated site of action [55]. With emerging multi-functional NPs, an additional selection of elements with unique properties is added, such as near-infrared (NIR) and MDP for photothermal and immune properties, respectively. The multiple variables playing a role in the synthesis of NPs warrant the development of unique nanoparticles. In return, the focus can easily be weighted toward novel research rather than supporting research, which improves our understanding of already designed nanoparticles and the potential for enhancing their properties. Figure 4 illustrates the ideal properties of a NP for drug delivery [88].

### 5.3. Future Perspectives

Oncolytic virus (OV) therapy is an emerging method of therapeutics consisting of modifying viruses and transfecting them into tumor cells, with some directly killing the cell while others being able to trigger the host’s immune system to fight the cancerous cells, with the potential of treating chemoresistant RB. Popular OVs include herpes virus and adenovirus [89]. Pascual-Pasto (2019) et al. studied the oncolytic adenovirus VCN-01 in targeting chemoresistant RB. This OV targets the altered RB1-E2F pathway causing cancerous cells to dysregulate the cell cycle, leading to uncontrolled proliferation. Through intravitreal injections in mouse Y79 xenograft models, this method induced tumor necrosis and ocular survival and increasing survival following a single intravitreal dose (*p* = 0.0002). Despite this, 2 out of 12 treated mice had detectable viral genome in the plasma 4 h after injection and in the brain tissue. Low systemic dissemination in the liver of rabbit models at day 3 was also observed. Clinical studies on two patients with RB, which was chemo- and radiotherapy-resistant, demonstrated intravitreal inflammation, leading to one participant undergoing enucleation. The second patient demonstrated anti-tumor response, with a reduction in the size and amount of tumoral vitreous seeds, although the size of retinal tumor remained unchanged [90]. Targeting nanoparticles may be able to improve the therapeutic capacity of these OVs and diminish systemic complications, bridging the barrier between OV and RB therapy. Likewise, gene therapy is an emerging therapeutic modality, although the delivery of vectors remains a barrier due to the negative charge, high molecular weight, and hydrophilicity of non-viral vectors in particular [91]. Therefore, Gao et al. (2023) developed an eye drop containing polyamidoamine (PAMAM)—a polyplex—embedded with small interference RNA (siRNA) to achieve gene silencing in orthotopic WERI-RB-1 nude mouse models and overcome the challenges of vector delivery to the posterior segment of the eye. Q8W and N9W-penetratin (89WP)—an absorption enhancer—were added to enhance the efficacy of topical delivery. The spherical particles were 200 nm in size, with a Zeta potential of 30 mV. In vitro experimentation on ARPE-19 and WERI-RB-1 cells highlighted the difficult internalization of free siRNA due to its high molecular weight and other characteristics. Use of PAMAM as a vehicle allowed for 3% and 20% uptake by both cell lines, respectively. The addition of 89WP increased the uptake in WERI-RB-1 cells by 90%. In vivo studies showed similar trends, namely that the polyplex reached the retina 2 h after treatment and remained there for 6 h, spotlighting the benefit of 89WP PAMAM in ocular delivery of non-viral vectors [92]. Their findings pave the way for analysis of various types of nanoparticles (e.g., gold, iron, mesoporous silica NPs) for delivering gene therapy to find appropriate NPs, which can effectively package, load, and compress siRNA [93]. Specifically, nucleic-acid-based NPs are also a point of discussion. Although not explored extensively in retinoblastoma, Wang et al. (2024) recently experimented with spherical nucleic acid (SNA) NPs to deliver siRNA. These SNA NPs with a dense DNA shell contain aptamers, targeting disialoganglioside (GD2) positive RB cells, and are loaded with siRNA and the inhibitor NVP-CGM097. These structures were 20 nm in diameter, and their loading efficiency ranged from 38.09 to 39.76%. The SNA NPs were also stable at room temperature for over 72 h. The in vitro binding capacity in Y79 and WERI-RB-1 cells was 89.2 and 87.4%, and the synergistic effect of siRNA targeting mouse double minute 2 homolog (MDM2)-mRNA transcription and NVP-CGM097 targeting MDM2 resulted in an inhibitory effect of 80.76% in Y79 cells, which was less than melphalan at 90.37%. However, in vivo studies on Y79 and WERI-RB-1 xenograft nude mouse models demonstrated that the SNA NPs caused less retinal damage than melphalan. Retinal thickness was noticeably reduced by melphalan, and the thickness only recovered on day 21 compared to a milder reduction and rapid recovery on day 14 in the SNA NP groups [94]. Compared to siRNA, DNA therapeutic agents are more stable and have a higher degradation resistance, making them potential candidates for RB treatment [95]. On the topic of nucleic acid NPs, Hu et al. (2024) developed aptamer-modified manganese dioxide (MnO_2_) nanosponge cores (hMNs), absorbing two DNAzyme agents. Aptamer was added for targeting properties toward nucleolin-overexpressed RB, shown by the increased uptake of the NPs in Y79 cells in vitro compared to healthy APRE-19 cells. The NPs were spherical, with a diameter of 200 nm and Zeta potential of −17.50 ± 2.00 mV. In vivo injection into Y79 xenograft nude mouse tumors illustrated the enhanced effect of aptamer. On day 14, the tumor volume of aptamer-modified NPs was 210.80 ± 17.84 mm^3^ compared to 338.68 ± 58.17 mm^3^ in the NP group without aptamer and 565.95 ± 47.63 mm^3^ in the control group [96]. To conclude, there is potential for DNA in aiding the construction of nanoparticles and for DNAzyme therapy, but there remains a need for concrete experiments evaluating the aforementioned DNA and siRNA. Additionally, our understanding of the role of DNA in engineering NPs for retinoblastoma remains unclear, with few studies focusing on its potential effect.

Moving away from therapeutics, the conventional monitoring of disease consists of a mix of fundus and histological examinations, and imaging, among others, which are limited by cost and resources. Liu et al. synthesized NPs using trisodium citrate dihydrate and ferric chloride hexahydrate for an RB monitoring platform via machine learning of aqueous humor metabolic fingerprinting using NP-enhanced laser desorption/ionization, demonstrating the potential of NPs beyond therapeutic application in retinoblastoma [97]. 

Table 2 summarizes the characteristics, features, and stage of research of the nanoparticles discussed in Section 4 and Section 5.

## 6. Conclusions

This review highlights the significant challenges in treating pediatric retinoblastoma and the emerging potential of nanotechnology in revolutionizing ocular drug delivery. Retinoblastoma, a leading intraocular cancer in children, presents challenges, such as anatomical barriers, drug instability, the risk of secondary tumors, and systemic toxicity from conventional treatments, such as chemotherapy and radiotherapy. Nanosystems, including gold, mesoporous silica, polymeric, and lipid nanoparticles, offer promising in vitro and in vivo solutions in animal models. Through these studies, it is clear that in order to target RB cells, the nanoparticles need to be small in size, stable, with a high drug loading capacity, and non-toxic to non-cancerous cells. Some of the studies discussed have also highlighted the preferential cationic property of NPs in favoring movement along the vitreous environment; this is particularly important in the context of retinoblastoma, as the most effective site of action is found behind the vitreous. Moreover, it is beneficial for these nanoparticles to be highly modifiable, allowing for targeted delivery. This can be achieved in many ways, including NP surface receptors targeting upregulated elements within tumor cells or pH sensitivity to tumor microenvironment. Although the addition of functional elements to a NP, such as PTT and imaging, can be beneficial, they are not necessary at the moment. Focusing on engineering simple mono-functional nanoparticles with the goal of treating retinoblastoma, enhancing current in vivo models to simulate the microenvironment of the eye, allowing long-term toxicity follow-ups, and then undergoing clinical trials to further their advancement are the next steps in the near future.

Despite its potential, the application of nanotechnology in treating retinoblastoma is still in its early stages, with hurdles such as long-term toxicity and limited clinical data. Future research is crucial to overcoming these barriers, improving simple nanoparticle design, and advancement toward clinical trials. By addressing these challenges, nanotechnology holds the promise of transforming retinoblastoma treatment, enhancing patient outcomes, and preserving vision with minimal adverse effects.

## Figures and Tables

**Figure 1 molecules-29-02263-f001:**
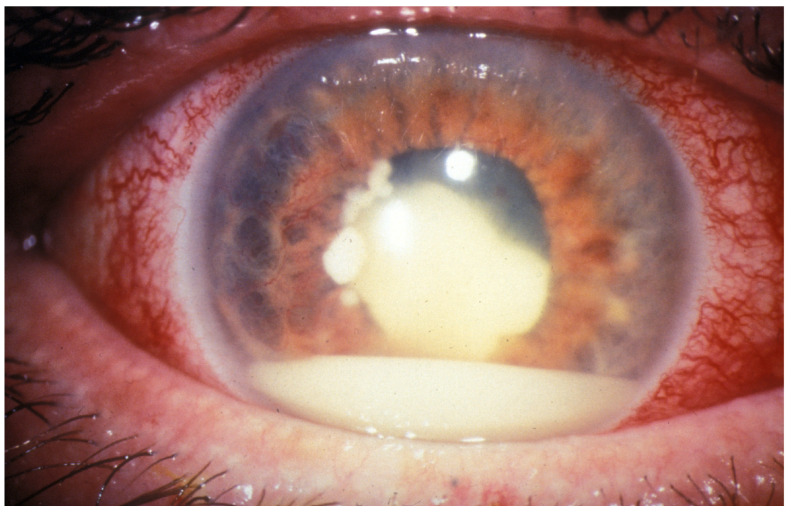
Manifestation of retinoblastoma accompanied by ocular inflammation and pseudo-hypopyon in the anterior chamber. Image credit: Prof. Clare Gilbert, International Centre for Eye Health (ICEH), London School of Hygiene & Tropical Medicine (www.iceh.org.uk, accessed on 24 February 2024). This work is licensed under a Creative Commons Attribution-NonCommercial-NoDerivatives 4.0 International License.

**Figure 2 molecules-29-02263-f002:**
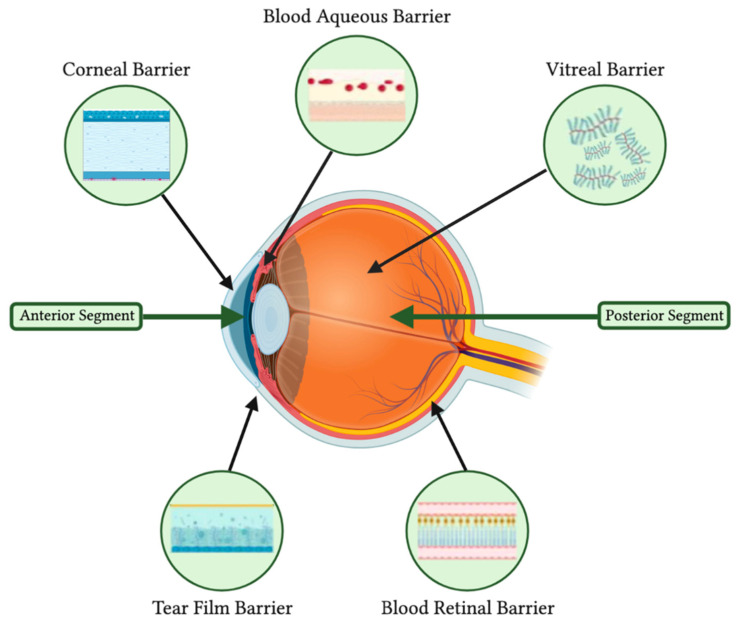
Anatomical and physiological barriers of the eye.

**Figure 3 molecules-29-02263-f003:**
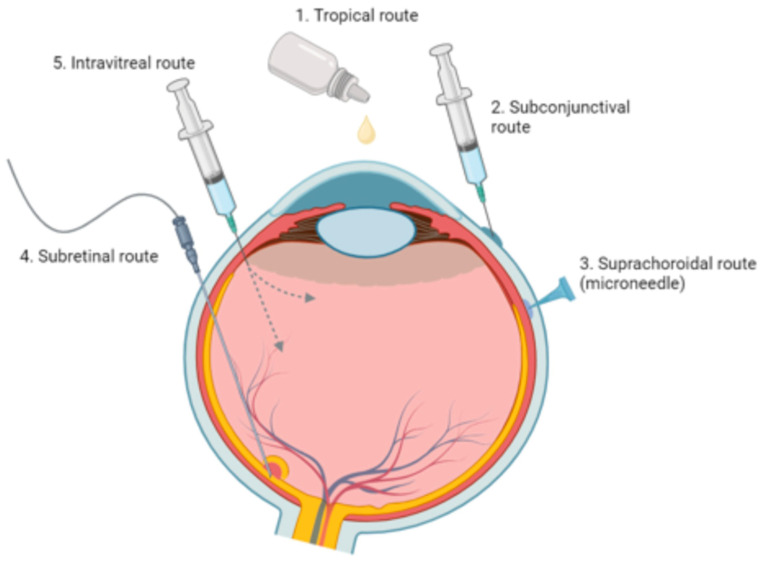
Ophthalmic medication delivery routes. This illustration features five modes of delivery, including topical, subconjunctival, suprachoroidal, subretinal, and intravitreal methods.

**Figure 4 molecules-29-02263-f004:**
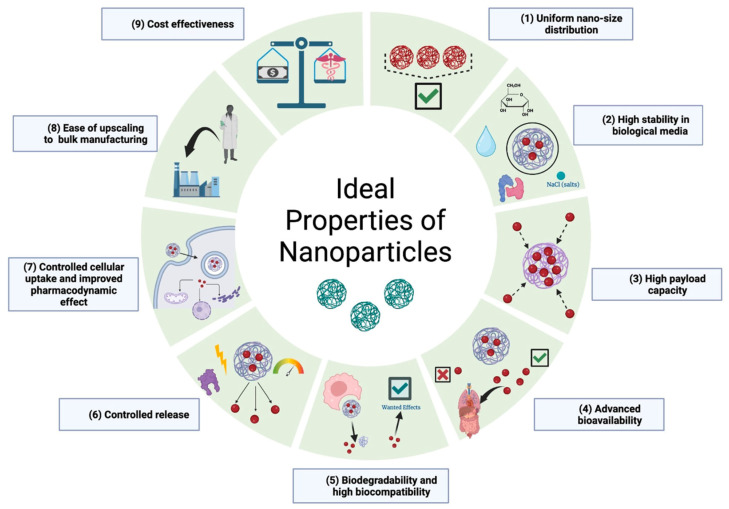
Ideal characteristics of nanostructures as drug delivery systems.

**Table 1 molecules-29-02263-t001:** Retinoblastoma—Current treatment challenges.

Treatment	Modality	Challenges	References
EBRT	N/A	- Secondary tumors - Treatment of last resort	[14,15,16,17]
Chemotherapy	Systemic	- Ineffective alone - Systemic toxicity	[18]
	Intra-arterial	- Invasive therapy - Need for equipment and specialists - Vascular side effects among others - Limited by BRB and avascular vitreous	[19,20,21,22,23]
	Intravitreal	- Toxicity to eye and surrounding tissue - Requires repeated injections, endophthalmitis	[24]

**Table 2 molecules-29-02263-t002:** Nanoparticles in retinoblastoma.

Nanocarrier	Function	Characteristics	Observations	Stage	Reference
GNPs	Target MDMX degradation	- Peptide-derived MG loaded - PSI coated	- PSI optimized blood circulation time - PH-dependent internalization optimal in cancer cells - Clearable through mononuclear phagocytic system - Absence of acute toxicity, no changes in weight, biochemical index of blood, or pathological findings	WERI-RB-1 cells in vitro, intravitreal orthotopic xenograft retinoblastoma mouse model in vivo, PDX model of pancreatic carcinoma in vivo	[27]
	Inhibit p53-HDM2	- Anti-HDM2 peptide loaded	- Cell arrest at G2M phase and increased apoptosis - Optimal cell internalization - NPs increased the stability and avoided degradation of the peptide	Y79 cells in vitro	[28]
	Laser-induced cell disruption	N/A	- NPs bubble when exposed to heat, causing cell membrane disruption - In vitro NPs attachment to cells is heterogenous - Vitreous body advantageous for treatment - Absent regeneration at 40 J cm^−2^ or more - Successful irradiation of cells and cell aggregates	Y79 cells in vitro and vitreous phantom	[29]
	Ultrasonic hyperthermia	N/A	- Ultrasonic waves allow for focused targeting - Cytotoxicity is proportional to hyperthermia time - Acoustic cavitation effect - Synergistic effect present	Y79 cells in vitro	[30]
	Ultrasound hyperthermia	- Evaluation of the synergistic effect of brachytherapy, ultrasound hyperthermia, and gold nanoparticles	- Combination group showed greatest inhibition of tumor growth - Combination group showed more adverse effects - Retinal gliosis and scarring were found - Need for therapeutic planning	RB-induced rabbit cells in vivo	[31]
	Imaging, LIFU, and immunotherapy	- MDP, PFP loaded	- Fe_3_O_4_ magnetic and T2 enhancing properties - MDPs promote Dc maturation and recognition of tumor cells - Biosafe and compatible in vitro and in vivo	Y79 and ARPE-19 cells in vivo, Y79-tumor-bearing mice in flanks in vivo	[85]
MSNPs	Enhance topotecan	- Topotecan loaded - Folic acid conjugated	- Folic-acid conjugate allows for increased uptake in RB cells - Enhanced anti-cancer effects due to increased uptake in vitro and in vivo - Controlled drug release	Y79 cells in vitro, Y79-cell-bearing xenograft tumor model in vivo	[34]
	Target malignant cells	- Carboplatin loaded - EpCAM conjugated	- Sustained drug release - Endocytosis pathway - Higher anti-cancer effect and apoptosis	Y79 in vitro	[36]
	N/A	- Glucuronic acid conjugated - Polyamidoamine dendrimer conjugated - Deferasirox loaded	- PH-dependent release - High loading efficiency and controlled delivery - Sugar-modified surface allows for improved cellular uptake	Y79 in vitro	[37]
Iron NPs	Magnetic hyperthermia using iron NPs	- Dextran coated	- Minimal cytotoxicity of NPs in absence of magnetic hyperthermia - Apoptosis through intrinsic and extrinsic pathways - Diffuse in Y79 cells but localized to lysosomes and endosomes in ARPE-19 cells	Y79 and ARPE-19 cells in vitro	[39]
SPIONs	Target mitochondria	N/A	- Decreased SDH activity - Increased MMP and mitochondrial swelling - SPIONs decline cell viability and increase caspase-3 activity	Y79 cells in vitro	[40]
	Chemo-hyperthermia therapy	- VCR loaded - FA and TF conjugated - Pluronic coated	- Toxicity in Y79 higher, indicating receptor targeting - NPs enhance ROS production upon AMF - Increased half-life compared to free drug - Low toxicity and good biocompatibility - Magnetic	Y79 and ARPE-19 cells in vitro	[78]
AgNPs	N/A	- Laminarin loaded	- Cytotoxicity in a dose-dependent manner - IC50 16.45 μg/mL - Apoptosis through extensive double strand breaks - Good scavenging activity, dependent on concentration of the AgNPs	Y79 cells in vitro	[47]
	N/A	- Laminarin loaded	- IC_50_ 10.5 μg/mL - Cytotoxicity depends on size and dose concentration - Causes morphological cell changes - Promising free radical scavenging effect	Y79 cells in vitro	[46]
PLGA NPs	Treatment with galactose conjugated ENPs	- Galactose conjugated - Chitosan coated - Etoposide loaded	- Sustained release up to 32 h - Greater cytotoxicity and apoptosis compared to free drug - Toxicity against HepG2 and Caco-2 reduced once loaded - Pure PLGA NPs non-toxic	Y79 cells in vitro	[51]
	Evaluate difference in NP loading vs. free	- OA and UA loaded	- NPs reduce drug toxicity at high concentrations - NPs only display significant cytotoxicity in Y79 cells - Lower IC50 in free drugs as compared to NPs	Y79, HepG2, Caco-2 cells in vitro	[52]
	Effects of these NPs on RB	- Carboplatin loaded - Sodium alginate surface modified	- Slow and sustained delivery with presence of SA - SA allowed for increased cellular uptake for unknown reasons - Intraocular toxicity is unknown - Potential for topical delivery	Y79 cells	[53]
PLGA NPs	Compare various surface-modified PLGAs	- Melphalan loaded - MGP, PEG, TET1, Avidin-Palm surface modified	- Adapted synthesis of PLGA NPs allowed for increased drug loading - Efficacy based on drug and NP concentration - Surface-modified MPG showed similar efficacy to free melphalan - Cell internalization increases with time, but a significant number of NPs associated with cell surface	Y79 cells in vitro	[23]
PCL NPs	Photothermal potential	- MO and NIR loaded	- PTT efficiency proportional to intensity of NIR light - NIR light facilitates NP release - Imaging properties of MO - Synergistic effect via downregulation of HSP70	Y79 cells in vitro	[84]
TMC and TC Nps	Biodistribution of intravitreal injection	- Thiole and methyl conjugated - CMD loaded	- Higher affinity of CMD-TCs-NPs for retina - TMS-derived NPs are immobilized in vitreous	Y79 cells in vitro	[55]
	Delivery of TPH	- TPH loaded	- Drug release GSH-sensitive in vitro - Increased cytotoxicity after loading - TPH-TCs-NPs show highest cytotoxicity due to thiol group percentage in vitro and in vivo - Enhanced cellular necrosis by TPH-TCs-NPs	BSS and FBS media in vitro, in vitro RB xenograft mouse model in vivo	[54]
TC-NPs	Intravitreal delivery	- TPH and CMD loaded	- Improved apoptosis compared to free drug - Suppression of RB tumor growth in NP treated rabbits - No retinal damage observed on histology	Y79 cells in vitro, xenograft RB rabbit models in vivo	[56]
SLN	Posterior eye delivery optimization	- Etoposide loaded	- Particle size, PDI, % entrapment efficiency affect therapeutic potential of SLN - Biphasic release with initial burst, then sustained release - Reduced toxicity of retina and blood plasma concentration compared to free drug	In vitro and rat eye in vivo	[58]
NLC	Stable delivery of PTX	- PTX loaded - α-tocopherol succinate based - PEG conjugated	- Maximum drug release of 40% - No difference in properties following sterilization - Properties of lipids and nanoparticles affect therapeutic properties - PEG increased physical stability of NPs	In vitro	[59]
LNPs	PTT	- BPQDs and melphalan loaded	- Upregulates ki67 expression - Cellular uptake within 20 min - Early and increased apoptosis - BPQDs-based NPs inhibit cell proliferation and killing - Photothermal efficiency related to concentration power and time - LNP reduces cytotoxicity of free BPQDs	RB cells in vitro, orthotopic xenograft of WERIRB-1 cells in mice in vivo	[63]
Switchable LNPs	Combination therapy	- Melphalan and miR-181a loaded	- Increased bioavailability of drug following encapsulation - MiR-181a decreases MAPK1 and Bcl-2 and enhances BAX - Combined loading has enhanced effects compared to co-delivery of separate entities - Switchable lipids change conformation at acidic pH	Primary RB cells and Y79 in vitro, xenograft RB models in rat in vivo	[60]
	Sensitize cancer cells to chemotherapy	- siRNA loaded	- Switchable lipids change conformation at acidic pH - Survivin silencing reduces therapeutic dose of drugs, except teniposide - siRNA sequence specific to survivin - Potential benefits of using drug-loaded LNPs instead of free drug after silencing for enhanced effects	13 cell lines, including Y79 RB cells in vitro	[61]
LFNPs	Evaluate efficacy of novel NP in chemoresistance	- CPT or EPT loaded	- Concentration of free drugs higher than drug-loaded NPs - Cytotoxicity of drug-loaded NPs higher than free drug - NPs increase drug uptake, retention	Y79 CSCs in vitro	[65]
Nanomicelles	Evaluate the potential of CDF in RB therapy	- CDF loaded - Folic-acid conjugated	- Selective cytotoxicity to RB cells compared to ARPE-19 cells - Nanomicelles in absence of drug show insignificant toxicity - Free CDF faces similar hurdles as curcumin in terms of solubility and degradation - Targeting NPs showed lower IC_50_ compared to non-targeting counterparts	Y79, WERI-RB, ARPE-19 cells in vitro	[72]
CNMs	Evaluate Celastrol potential in angiogenesis	- Celastrol loaded	- CNMs inhibit HUVECs migration in dose-dependent manner - Blocking VEGF-A pathway - Decrease tumor volume and weight - No influence on mouse body weight - Histology staining shows decreased vascularization	HUVECs and SO-RB 50 in vitro and xenograft mouse model in vivo	[74]
RSNMs	N/A	- Celastrol loaded	- Activation of caspase-related proteins - Cellular uptake seen in under 10 min - Slow diffusion from cell to its nuclear contents - Apoptosis was time-dependent	Y79 cells in vitro	[75]
CMT NPs	Dual-mode image-guided laser/immune co-therapy	- Tuftsin coated	- Good thermal conversion ability - Low cost compared to gold NPs - Tuftsin is macrophage-specific - Concentration of NPs increases under magnetic field - PA signals remained after 24 h; potential in image-guided tumor therapy - Negative enhancement of T2WI MRI imaging - Successful PPT in vivo, decreasing tumor volume - Tuftsin induces macrophage typing - Absence of toxicity using CMT	Y79, ARPE-19, Ana-1 in vitro, Y79-tumor-bearing mice in vivo	[80]
LNPs	Dual-target cationic	- Folate conjugated - Spions, PFH, ICG loaded	- Folate magnetic dual target increased cellular uptake - Synergistic effect of ICG on PDT and PTT - Excellent biosafety - PFH allows for oxygen reserve, enhancing PDT effect	Y79 and ARPE-19 cells in vitro, Y79-tumor-bearing mice in vivo	[81]
	Multi-functional LNPs	- Folate conjugated - DOX, ICG, PFP loaded	- PFP enhances drug release through microbubbles induced via laser irradiation - FA-DOX-ICG-PFP-LNPs +laser irradiation showed the lowest cell viability and synergistic effect - Strong intake of NPs in liver and spleen, no histological abnormalities	Y79 and HUVECs in vitro, xenograft mouse models in vivo	[86]
	Multi-functionality, including gene therapy	- FA conjugated - PFP, ICG, TK-GFP plasmid loaded	- PA/US contrast following irradiation - Synergistic effect of electrostatic absorption and folate targeting on internalization - Cationic property of NPs allows for plasmid binding - Subcutaneous mouse tissue in vivo rather than eye	Y79 cells in vitro, nude mouse limbs in vivo	[83]
PNPs	PCB and PTT combination therapy	- PLGA and PCL embedded - PCB and NIR loaded	- Time-dependent increase in temperature allowing for controlled drug release - Adequate photoacoustic imaging properties - Synergistic cytotoxicity - Absence of behavioral toxicity, mortality, or body weight decrease	Y79 cells in vivo, mouse models in vivo	[82]
Clathrin-like NPs	Overcome BRB	- Dox loaded	- Cellular uptake increased following light irradiation in vitro and in vivo - Minimal eye irritation and no structural damage - Unnoticeable systemic toxicity - Free drug treatment caused body weight decrease of 10%, whereas absent in loaded NPs	HUVECs and WERI-RB-1 cells in vitro, and orthotopic retinoblastoma tumor mouse model in vivo	[76]
N/A	VCN-01 therapy	N/A	- VCN-01 cytotoxic against RB cells in vitro - Intravitreal injection, slight and short-term leakage into blood - 2/12 mice had detectable viral genome in plasma 4 h after injection - Clinically feasible with anti-cancer effects on vitreous seeds - Local vitreous inflammation	Y79 and primary RB cultures in vitro, mouse, rabbit, and human in vivo	[90]
PAMAM	Non-viral gene therapy	- siRNA loaded	- Enters cells through clathrin-mediated endocytosis - Topical administration reaches the posterior segment of the eye within 2 h - Elimination within 6 h	ARPE-19 and WERI-RB-1 cells in vitro	[92]
SNA NPs	siRNA delivery	- siRNA, NVP-CGM097, aptamer loaded	- Synergistic effect of loaded components on MDM2 pathway - Melphalan showed in vitro superiority in cell killing - Both SNA NPs and melphalan reduced pupil leukosis, a sign of anti-tumor effect - Melphalan exerted more toxic effect on retinal thickness compared to NPs	Y79 and WERI-RB-1 cells in vitro and in vivo in nude mouse xenograft models	[94]
MNO_2_ and DNAzymes NPs	Dual-gene therapy	Aptamer surface modified	- DNAzymes target two separate mRNAs - Effective delivery to cells within 4 h - Potential for dual-mode imaging - Aptamer increases cellular uptake in vitro and in vivo - Absence of toxicity in vivo due to absent body weight change - Increased H&E staining tumor cell necrosis compared to control	ARPE-19, Y79 cells, Y79-tumor-bearing mice	[96]

## Data Availability

Not applicable.

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
