# Peer review of "Advancements in Nanosystems for Ocular Drug Delivery: A Focus on Pediatric Retinoblastoma"

_molecules, 2024, doi:10.3390/molecules29102263_

Round 1
Reviewer 1 Report
Comments and Suggestions for Authors
· Based on the review of literature the authors should identify the ideal features of NPs used in ocular delivery and especially in the treatment of retinoblastoma
· It is confusing why the authors did not use the conventional classification of inorganic NPs to describe the gold, mesoporous silica, iron NPs etc and decided to include chitosan and silver NPs under a different category.
· For each of the different type of NPs it is worth highlighting the specific features of these materials that make them ideal for this delivery strategy
· The authors should provide sufficiently more in depth detail about each of the studies highlighted as the information provided in certain sections about some of the studies were too broad and non-specific – for example in order to provide the reader with sufficient information about the
o particle size, drug loading efficiency, zeta potential, in vitro drug release profile of the NPs highlighted
o specific details of any in vitro, ex-vivo or in vivo studies such as the models or animals used in testing and summarised data for example on Page 12 line 472 – increased cellular uptake of the formulation was highlighted but may improve the submission if actual information of how much increase is included for the reader to appreciate the impact of some of the studies. This goes for majority of the studies reviewed. A good example of some further detail was on Page 7 line 238 with actual values supporting the statement. Therefore, the authors should review the submission
· Lots of abbreviations that needs to be defined, inconsistencies in use of abbreviations
· Inconsistences in in text referencing format – some in author date format some just author
· Authors have included some studies that were out of range of the years identified for this review
· Review numbering of figures which should also be referred to in the text
Author Response
Dear Reviewer,
Thank you for your thoughtful feedback on our manuscript. We have carefully considered each of your comments and made revisions to address them comprehensively. Here is a detailed response to your specific concerns:
Ideal Features of Nanoparticles (NPs) for Ocular Delivery:
We have emphasized the ideal features of nanoparticles used in ocular delivery, particularly in treating retinoblastoma, in the conclusion section. Your suggestion was valuable, and we've highlighted key properties that are crucial for effective treatment.
Classification of Inorganic NPs:
We understand the confusion arising from our previous classification. The categorization has now been updated to distinguish between inorganic and organic nanoparticles. We have clearly placed gold, mesoporous silica, and iron nanoparticles under inorganic, while categorizing chitosan and silver nanoparticles separately.
Specific Features of NPs for Delivery:
Before each NP type (e.g., gold, silver, iron), we have included a concise explanation of their unique properties that make them suitable for ocular drug delivery. We've also emphasized why specific experiments were conducted and how certain barriers are being addressed. Your input has helped us achieve greater clarity in this regard.
In-Depth Study Details:
We provided additional information on particle size, drug loading efficiency, zeta potential, and drug release profiles of nanoparticles to give a clearer picture. We've also expanded on the details of in vitro, ex-vivo, and in vivo studies, including models or animals used, and offered specific numerical results when possible. For instance, the increased cellular uptake mentioned on Page 12, Line 472, now includes exact values, providing a quantitative understanding.
Abbreviations and Inconsistencies:
All abbreviations have been reviewed and defined consistently. Any inconsistencies in the use of abbreviations have been rectified.
In-Text Referencing Format:
Inconsistencies in referencing have been corrected, ensuring all references follow the same author-date format.
Studies Out of the Year Range:
The statement has been removed. Some studies that fell outside the specified timeframe for this review have been removed.
Figure Numbering:
We reviewed the numbering of figures and ensured that all figures are now correctly numbered and referenced in the text.
We sincerely appreciate your constructive comments, which have significantly improved the quality of our manuscript. Please feel free to reach out with any further suggestions or concerns.
Best regards,
Reviewer 2 Report
Comments and Suggestions for Authors
The review by Kevin Y. Wu and colleagues provides a comprehensive overview of the advancements in nanosystems for ocular drug delivery particularly focusing on pediatric retinoblastoma. The main question focused on by the review article is how nanotechnology can improve drug delivery for retinoblastoma treatment.
The review highlights various types of nanoparticles being investigated for retinoblastoma treatment, including Gold nanoparticles, Mesoporous silica nanoparticles, Iron nanoparticles, Polymeric nanoparticles, Lipid nanoparticles, Protein nanoparticles, Nanomicelles, Silver nanoparticles, etc. The review focuses on the potential of various nanoparticles (as mentioned above) for retinoblastoma treatment. It highlights the gap in knowledge regarding the long-term effects and safety of these nanoparticles while acknowledging the lack of extensive clinical data.
The reference list is comprehensive and includes relevant literature between 2017 and 2023.
The authors should mention the full forms for the first time throughout the review, e.g., EBRT.
The language is clear and professional throughout the review.
Figure 1 is presented twice in the review. The authors need to change the figure numbers and figure explanations accordingly throughout the review.
In Figure 1 on Page 6, the authors need to increase the font size of the text. The same changes are also needed for Figures 2 and 3.
In Table 2, nanoparticles in retinoblastoma, the authors should check the reference mentioned for PTEB micelle. What is (4/1/2024 1:38:00 PM)?
As the authors have discussed various types of nanoparticles in this review, it would be helpful if they could comment on DNA-based nanoparticles in retinoblastoma.
Overall this review provides a comprehensive overview of different nanoparticles being explored for retinoblastoma treatment, including their advantages and limitations. It emphasizes the need for future research on improving nanoparticle design, addressing long-term toxicity concerns, and conducting clinical trials.
Author Response
Dear Reviewer,
Thank you for your comprehensive review and valuable feedback on our manuscript. We've carefully considered your comments and made revisions to address them. Here is a point-by-point response to your specific suggestions:
- Abbreviations: We have ensured that all abbreviations, such as EBRT, are fully spelled out when first mentioned. Your suggestion has helped us make the text clearer and more accessible to all readers.
- Duplicate Figure 1 and Figure Numbering: We identified and resolved the duplicate instance of Figure 1. We have also updated figure numbers and explanations throughout the review, ensuring that all figures are appropriately numbered and referred to in the text.
- Figure Font Size: We addressed the legibility concerns by editing the images. This solution should help improve the clarity of the text in Figures 1, 2, and 3.
- PTEB Micelle Reference in Table 2: we have addressed this issue by carefully reviewing the table and removing the inadequate entry ("4/1/2024 1:38:00 PM").
- DNA-Based Nanoparticles: We included a brief commentary on DNA-based nanoparticles in the "Future Perspectives" section. However, the available literature on this subject is limited, and we could only provide a succinct overview. Nevertheless, your suggestion was valuable in encouraging us to mention this important but underexplored topic.
Overall, your insightful comments have helped us strengthen this review. We hope the revisions made align with your expectations. Please do not hesitate to reach out with any additional suggestions or clarifications.
Best regards,
Round 2
Reviewer 1 Report
Comments and Suggestions for Authors
Thanks for addressing all the comments raised in the previous review